# Exploring Transfer Potentials of the IMPROVE*job* Intervention for Strengthening Workplace Health Management in Micro-, Small-, and Medium-Sized Enterprises in Germany: A Qualitative Study

**DOI:** 10.3390/ijerph20054067

**Published:** 2023-02-24

**Authors:** Anke Wagner, Brigitte Werners, Claudia Pieper, Anna-Lisa Eilerts, Tanja Seifried-Dübon, Matthias Grot, Florian Junne, Birgitta M. Weltermann, Monika A. Rieger, Esther Rind

**Affiliations:** 1Institute of Occupational and Social Medicine and Health Services Research, University Hospital Tübingen, Wilhelmstr. 27, 72074 Tübingen, Germany; 2Operations Research, Institute of Management, Ruhr-University Bochum, Universitätsstr. 150, 44801 Bochum, Germany; 3Institute for Medical Informatics, Biometry and Epidemiology, University Hospital Essen, University of Duisburg-Essen, Hufelandstr. 55, 45147 Essen, Germany; 4Department of Psychosomatic Medicine and Psychotherapy, University Hospital Tübingen, Osianderstr. 5, 72076 Tübingen, Germany; 5Institute of General Practice and Family Medicine, University Hospital Bonn, Venusberg-Campus. 1, 53127 Bonn, Germany

**Keywords:** workplace health management (WHM), job satisfaction, psychosocial stressors, micro- and small-sized enterprises (MSE), small- and medium-sized enterprises (SME), focus group discussion, single interviews, rapid data analysis, Germany

## Abstract

Micro- and small-sized enterprises (MSE), and small- and medium-sized enterprises (SME) in Germany are often burdened with high levels of psychosocial stressors at work. The IMPROVE*job* intervention was originally developed for general practice teams, and aims to promote job satisfaction and reduce psychosocial stressors in the context of workplace health management (WHM). This qualitative study identified challenges and transfer options regarding the transfer of the IMPROVE*job* intervention to other MSE/SME settings. Based on previous study results, a comprehensive, qualitative inter- and transdisciplinary approach was developed and conducted between July 2020 and June 2021, also including single interviews and focus group discussion with eleven experts from MSE/SME settings. Data analysis was carried out using a rapid analysis approach. The experts discussed psychosocial topics and didactic formats of the original IMPROVE*job* intervention. A lack of access to information on managing work-related psychosocial stressors and inadequate recognition of the importance of psychosocial stressors in the workplace among managers and employees, seemed to be the highest barriers regarding the transfer of the intervention into other MSE/SME settings. The transfer of the IMPROVE*job* intervention to other MSE/SME settings requires an adapted intervention format, comprising targeted offers with easy access to information on managing work-related psychosocial stressors and improving WHM in MSE/SME settings.

## 1. Introduction

Every workplace has to deal with a variety of work-related psychosocial stressors in terms of leadership [1], work content, organization of work, social relations, working environment, and new forms of work [2]. Legislation in Europe [3] and Germany [4] is based on the following principle: work must be organized in such a way that hazards to life and physical and mental health are avoided as far as possible, and the remaining hazards are kept as low as possible [4]. Workplace health management (WHM) aims to implement health-related measures in companies, with the aim of reducing work-related psychosocial stress, as well as maintaining and improving the health of employees [5]. Workplace health promotion is therefore an essential component of workplace health management. It encompasses the areas of health and safety at work and workplace integration management, as well as personnel and organizational policy. It includes all measures implemented in the company to strengthen health resources [6]. Previous studies have demonstrated that workplace health promotion measures are not widely implemented in smaller enterprises compared to larger companies [7,8,9,10]. According to the European Commission, micro- and small-sized enterprises (MSE) comprise typically less than 50 employees, whereas small- and medium-sized enterprises (SME) include less than 250 employees, and adhere to certain limits with regard to either an annual turnover or an annual balance sheet total [11].

Overall, it has become evident in recent years, that a comprehensive WHM is often not well implemented in MSE/SME settings [7,12,13], particularly in Germany. MSE/SME enterprises regularly have fewer employees, operations, structures, and equipment than larger companies [14]. Thus, it can be assumed that in MSE/SME settings there are generally fewer resources available to deal with, develop, and implement suitable approaches and interventions addressing psychosocial stressors at work [15]. Smaller enterprises need targeted support and advice, as well as an infrastructure, to implement activities related to WHM [8,9].

So far, only a few studies on interventions have been conducted, with the aim of developing suitable approaches for better handling of psychosocial stressors at work in MSE/SME settings [16,17,18,19,20,21]. Within the Business in Mind study, a complex intervention for the promotion of mental health was successfully developed and implemented in different SMEs [16,17,18]. The currently ongoing KMU-GO study (in German: Kleine und mittlere Unternehmen–Gesundheitsoffensive) aims to implement and evaluate stress management training for leaders/managers in SMEs in Germany [19]. The PragmatiKK study (in German: Pragmatische Lösungen für die Implementation von Maßnahmen zur Stressprävention in Kleinst- und Kleinbetrieben) developed strategies and solutions for the reduction in work-related stress in MSEs [20,21]. A recently conducted systematic review searched for and evaluated different workplace interventions for the reduction in anxiety and depression in SME settings [22]. The review identified few studies, with high heterogeneity, and concluded that further research in this setting is urgently needed [22].

Most of the interventions so far, focused on behavioral preventive approaches, and targeted leaders and managers [16,17,18,19]. In close cooperation with a research support group, consisting of general practitioners and practice assistants, and an advisory board, the multimodal participative IMPROVE*job* intervention was developed by a transdisciplinary research collaboration, comprising experts in the fields of general practice and family medicine, occupational medicine, psychosomatic medicine, operations research, and workplace health promotion [23,24,25]. Thus, the IMPROVE*job* intervention was implemented for substantial improvements in the context of WHM, and addresses the topic “mental health for managers and employees in primary care practices” considering aspects of behavioral and structural prevention [23,24]. The intervention aimed to enhance job satisfaction as well as reduce commonly perceived psychosocial stressors (e.g., high levels of work intensity, frequent interruptions) in general practice teams (GP) [24,26,27]. The main outcome of the intervention was job satisfaction, and the intervention addressed topics regarding leadership, teamwork, social relations, occupational health and safety, and workplace health promotion, as well as work organization and work process [23,28]. Using the primary care setting as an example for MSE/SME settings, the IMPROVE*job* intervention was evaluated regarding its effectiveness in a cluster randomized controlled trial (cRCT) in 60 GP practices, within an intervention and a control group, in 2019 and 2020 in the North Rhine region in Germany [29].

The IMPROVE*job* intervention consisted of the following parts: two leadership workshops—workshop 1 for practice leaders only, and workshop 2 for practice leaders and their teams—a toolbox with supplemental material, and the use of IMPROVE*job* facilitators [23,28,29,30]. A detailed overview of the components of the IMPROVE*job* intervention and their aim is provided in Table 1.

The study presented here is based on topics of, and experiences with, the IMPROVE*job* intervention outside the primary care setting [23,28]. In general, it seems promising, resource-saving, and more easily applicable for MSE/SME settings, to use an already developed intervention and adapt it for transfer into other work sectors. Therefore, we aim to empirically discuss and explore with different experts:(1)the relevance of psychosocial topics of the original IMPROVE*job* intervention in different MSE/SME settings;(2)the main challenges faced by owners and managers in different MSE/SME settings;(3)as well as, possible transfer options (barriers and facilitators) regarding the implementation of the IMPROVE*job* intervention into other MSE/SME settings.

## 2. Materials and Methods

### 2.1. Study Design and Preparation of the Field Work

A qualitative inter- and transdisciplinary research design was applied comprising (a) a continuous literature search for psychosocial stressors in different MSE/SME settings, (b) eight interdisciplinary working group meetings within the IMPROVE*job* research collaboration, to reflect and discuss the transferability of certain topics of the IMPROVE*job* intervention to other MSE/SME settings, (c) a transdisciplinary workshop, with an advisory board of experts either working in the MSE/SME sector or providing additional academic background from the areas of occupational psychology, WHM, mental health, and change management, and (d) single interviews and a transdisciplinary focus group discussion with experts from MSE/SME settings, to elaborate further on challenges and transfer options of the IMPROVE*job* intervention to other MSE/SME settings. The working group meetings took place regularly, to ensure a continuous exchange (July 2020–July 2021). The concept for the qualitative data collection was based on central topics of the IMPROVE*job* intervention: leadership and leadership role, including the appreciation and implementation of occupational health and safety (workshop 1), social relations and communication within the team and with patients, as well as aspects of work processes and work organization (workshop 2). Based on the discussions, the feedback, and the professional exchange within the project network and with the extended project advisory board, the concept for the qualitative data collection was finalized.

### 2.2. Recruitment and Qualitative Data Collection

Based on the preliminary work described, we initially planned to conduct three focus group discussions with different experts from MSE/SME settings. The experts were recruited through the “Offensive Mittelstand” (a national German initiative to promote quality of work in MSE/SMEs) and other work-related and personal networks of the cooperation partners. The Offensive Mittelstand is a practice partner of the Institute of Occupational and Social Medicine and Health Services Research. The recruitment strategy was developed and consented to by the IMPROVE*job* consortium, and then discussed in a transdisciplinary online meeting with leaders of member enterprises of the Offensive Mittelstand in February 2021. The partners of the Offensive Mittelstand gave practical advice, and received an information sheet on the study after the online meeting and were asked to share it in their networks during the recruitment period. They were also sent an appointment request to share, in which potential participants could sign up for one of the three focus group discussions. Unfortunately, the recruitment of experts from different MSE/SME settings turned out to be very difficult, and at the end of the recruitment period only seven participants for one focus group discussion were found. Therefore, the study team decided to slightly modify the concept for the study, and to conduct single interviews and one focus group discussion. We used the concept “information power” [31] to guide adequate sample size, with the aim of achieving a broader representation of the, already very specific, topics defined in the IMPROVE*job* intervention (see Table 1). As a rule of thumb; the more specific the individual dimensions, the fewer cases are needed.

The single interviews took place from the end of May 2021 to the beginning of June 2021. Five different experts from MSE/SME settings were asked to participate in single interviews. For the single interviews, the participants received an email in advance, with information on the course and the duration of the interview, information on the study, and explanations regarding the data protection concept, as well as a letter of consent. Three interviewers (A.W., C.P., and B.W.) with professional backgrounds in health services research, operations research, and workplace health promotion conducted two telephone interviews and three face-to-face interviews with five experts, using an interview guide with questions capturing attitudes regarding the following topics: job satisfaction, team communication, leadership, work organization, and work process (for more information on the interview guide see Appendix A). Each interview was conducted by one of the interviewers. The interviews took on average 29 min. One interview was recorded. In agreement with the other two interviewees, these interviews were not recorded, but the interviewers documented the main topics/issues using a template. A self-developed standardized template was filled out for each interview by summarizing the main responses to the questions (for more information on the template see Appendix A). The experts came from companies in the fields of legal services, production, construction-architecture, hairdressing, building trade, and business consulting, and had held leadership positions for an average of 16.8 years (range of 5–27 years). The companies were MSE and small enterprises, with three to twenty-two employees.

A digital transdisciplinary focus group discussion was scheduled, due to the ongoing COVID-19 pandemic. Prior to the start of the focus group, all participants received an invitation email with information about the focus group discussion (information about the process and the scheduled duration of 3 h, information on the study and the data protection concept, as well as a letter of consent). Additionally, information on the use of the ZOOM^®^ video telephony software was also sent (including link, meeting ID, and password). The focus group discussion took place at the end of June 2021 using the ZOOM^®^ software (Zoom Video Communications). Thereby, the data protection conditions of the license of the University of Tübingen were applied. Six participants took part; one person was unable to attend at short notice. The experts were from industry, employers’ associations, and university, with professional backgrounds in WHM, mental health, industrial and organizational psychology, occupational health and safety, vocational education and training, economics, change management, and development of organizations and employees. The experts had various insights into MSE and SME sectors and what special requirements exist in particular in these areas. In addition to the experts, three researchers from the IMPROVE*job* consortium (E.R., M.A.R., and A.W.), and one additional person (E.Ö.), attended the workshop. The focus group discussion was moderated by ER. Due to strict data protection regulations, the focus group discussion was not recorded, and only protocols of discussions were produced by two other persons (A.W. and E.Ö.), using a self-developed standardized template (for more information on the template see Appendix A). The two persons had a background in health services research and sociology, and were experienced in qualitative research. At the beginning of the focus group discussion, the participants received detailed information about the topics of the IMPROVE*job* intervention, and how these were implemented in the GP setting within the cluster randomized controlled trial. Then, in two discussion rounds, open questions were addressed about general leadership challenges (first discussion round) and possible transfer options (second discussion round) into other MSE/SME settings. During the focus group discussion, the experts often agreed and confirmed their attitudes on various discussion points, and the researchers could not detect any disagreements. The detailed schedule of the focus group discussion is presented in Appendix A. At the end of the focus group discussion, the participants had the opportunity to evaluate the meeting using a short standardized online questionnaire, by QuestBack Unipark^®^. The questionnaire was for assessment and evaluation of the workshop within the following six topics: organization of the workshop, promotion of mutual interaction between the participants, satisfaction with the selection of the topics, satisfaction with the given impulses for the discussion, assessment of the relevance of the chosen topics for the daily work, and overall satisfaction with the workshop. Most of the questions were based on a 5-point Likert scale (1 = disagree strongly/very dissatisfied; 5 = agree strongly/very satisfied). The items of the online questionnaire are shown in Appendix A.

### 2.3. Data Analysis

The data analysis of the interviews and the focus group was carried out using the qualitative rapid analysis method, following Miles et al. [32]. Data analysis according to this method involves three steps: (1) summarize the data, (2) prepare and present the data in a matrix (table), and (3) draw conclusions based on the data [32]. Data analysis was conducted separately for the focus group discussion and for the interviews, based on the two templates used during data collection. Due to the short timeframe of the study, it was not possible for the experts from the interviews and the workshop to receive the completed templates for checking and correction. For the data analysis of the interviews, a matrix was created for each topic (job satisfaction, team communication, leadership, work organization, and work process). After the interviews, the results were filled in by the interviewer and compared using the developed templates. For the data analysis of the focus group discussion, the two templates (one per recorder) were combined, and central statements of the experts on individual topics were summarized and mapped. The descriptive analysis of the standardized evaluation of the focus group discussion was performed with IBM SPSS^®^ Statistic^®^ Version 27, by calculating frequencies for the answers to each question. All results were then presented in an interdisciplinary work group meeting, and during the course of the discussion, implications for MSE/SME settings were jointly derived.

### 2.4. Ethical Considerations

The study was conducted according to the guidelines of the 1964 Declaration of Helsinki and its later amendments, or comparable ethical standards, as well as all data protection requirements. The study received ethical approval from the responsible Ethics Committee of the Medical Faculty and University Hospital of the University of Tübingen (project number: 173/2021BO2), and the data protection concept was also critically reviewed by the data protection officer of the University Hospital of Tübingen. All participants received a detailed information letter about the study, and were asked to sign a letter of informed consent in advance of the focus group discussion, or in advance of the five single interviews. It was possible to withdraw from participation in the study at any time.

## 3. Results

Overall, the results of the qualitative data collection are based on single interviews (*n* = 5), and a focus group discussion (*n* = 6) with experts from various MSE/SME settings. The experts in the single interviews evaluated and discussed psychosocial topics of the IMPROVE*job* intervention, whereas the experts in the focus group discussion elaborated various challenges and possible transfer options (see Figure 1).

Subsequently, the results of the single interviews and the insights from the focus group discussion (challenges and transfer options) are presented.

### 3.1. Psychosocial Topics of the IMPROVEjob Intervention

In five single interviews, the experts elaborated psychosocial topics of the IMPROVE*job* intervention. The five experts rated the topics, and also the didactic format of the IMPROVE*job* intervention, as highly suitable for the target group. In summary, they identified and suggested, for MSE/SME settings, different drivers for the promotion of job satisfaction, leadership, teamwork, and relationships, as well as work organization and work process. The results are presented in Table 2.

### 3.2. Challenges for Owners and Managers in MSE/SME Settings in Germany

Based on statements of the experts in the first discussion round of the focus group discussion, the current challenges for managers and owners in MSE/SME settings can be categorized into personal and general challenges.
Personal challenges comprise leadership issues (e.g., lack of leadership role understanding; little/hardly any time for leadership issues; leadership not seen as a responsibility; leadership tasks as a cost factor (no sales)/low priority; leadership of larger groups), work overload (e.g., multiple roles at the same time; supervision of many employees; and few resources), and possible knowledge gaps of managers and owners (e.g., little knowledge about mental stress and strain factors; attitude: “everyone is responsible for themselves”).General challenges include more emerging topics (e.g., demographic change and age-appropriate working design, dealing with diversity, and lifelong learning), organizational issues (e.g., vacation planning and sick leave; employee meetings; communication with customers; handling time management; cross-industry competition for skilled workers), team care and staff management (e.g., handling small teams; finding and retaining staff; handling low commitment to the company; different working attitude of the present generation; staff shortage; handling high stress and group dynamics; reaching all employees; equal treatment/fairness; communication under stress), and challenges regarding the implementation of changes (e.g., no facilities; no internal support; information overload; only general support; dealing with works council and management; no networking; no organizational structures in the company).

Therefore, as possible solution strategies, it was deduced that a broader commitment to leadership in MSEs/SMEs should be established among owners and managers, and that more knowledge is required for this purpose. Further, more awareness of the relevance of health and leadership measures should be created among owners and managers. According to the experts, required measures should be developed together with the team (“bottom-up approach”), and information about permanent, long-term measures should be provided for each employee.

### 3.3. Transfer Options Barriers and Facilitators for MSE/SME Settings in Germany

In the second discussion round of the focus group discussion, barriers regarding the transfer options of the IMPROVE*job* intervention into other settings were elaborated and discussed among the experts. Also, here, the experts of the focus group discussion gave a positive evaluation regarding the topics and the didactic format of the IMPROVE*job* intervention. The main two barriers for a successful transfer into other MSE/SME settings, according to the experts, comprised the following: (1) lack of access to MSE/SME settings to offer suitable approaches, and (2) that topics other than those chosen in the IMPROVE*job* intervention were considered as more important among owners, managers, and employees.

Based on the two identified barriers, several facilitators were then considered as appropriate to transfer the main topics of the IMPROVE*job* intervention to different MSE/SME settings. Facilitators mentioned in the discussion are shown in Table 3.

### 3.4. Evaluation of the Conducted Focus Group Discussion

The focus group discussion was evaluated with six close-ended questions. All six participants showed high levels of satisfaction regarding the conducted workshop. The organization of the workshop was rated very positively by five experts. Further, all six experts agreed that the workshop promoted mutual interaction and communication between the participants. They indicated that they were very satisfied (five experts) or satisfied (one expert) with the selection of the topics. Five experts believed that the given impulses in the two discussion rounds were helpful in initiating the discussion among the participants, and were also helpful to derive some ideas from the workshop for their day-to-day work. Overall, five experts were very satisfied, and one expert was satisfied with the conducted workshop.

## 4. Discussion

This qualitative study elaborated and discussed challenges and possible facilitators regarding the transfer of the multimodal participatory IMPROVE*job* intervention into other MSE/SME settings, with various experts on MSE/SME settings in Germany. All experts evaluated the topics of the IMPROVE*job* intervention as important and appropriate for different MSE/SMS settings. They further identified central drivers for the promotion of job satisfaction, teamwork and relationships, leadership, work organization, and work process in MSE/SME settings. In summary, the identified drivers reflect the special work culture and situation in MSE/SME settings. According to Cunningham et al., small businesses are often family run, and contain close relationships between the employer and employee that are not just work-related [14]. Thus, good social relations between these actors in MSE/SME settings are very important, and should be maintained. According to results of the joint analysis of the European survey of enterprises (ESENER-2 study), the required involvement of employees is crucial in this setting to successfully implement measures to minimize perceived psychosocial stress [15].

With the above-mentioned drivers, it is evident that MSE/SMEs could benefit from external consulting and support services. Thus, further approaches and interventions that specifically promote psychosocial topics of the IMPROVE*job* intervention in other MSE/SME settings are needed. The ESENER-2 study demonstrated that in the healthcare, education, and financial sectors, there are already many efforts to deal with psychosocial risk factors [15]. But in other sectors, such as mining, agriculture, and construction, few procedures and measures for the handling of psychosocial risk factors have been developed and implemented [15]. Thus, it can be assumed that owners, managers, and employees in these sectors could benefit particularly from further information on the relevance of work-related psychosocial issues.

The experts in the focus group discussion mentioned in the first discussion round various challenges with which owners and managers were confronted. The personal and general challenges described here are in line with results from other studies, and indicate that these challenges lead to higher perceived stress among owners and managers in MSE/SME settings. As elaborated by Lehmann et al., leaders in MSE/SME settings have multiple roles, broad responsibilities, and no, or only a few, co-workers on the same hierarchy level to exchange experiences or receive support [19]. According to Cunningham et al., owners and managers are responsible for several complex issues, and influence, with their personality, skills, and attitudes the success of their company [14]. Thus, they are aware of their central responsibility for their employees and for their company. In a study by Cocker et al., among a total sample of 217 owners and managers, 37% (*n* = 80) reported experiencing high levels of perceived stress [16]. High perceived stress among owners and managers correlated with higher absenteeism and presence at work when ill (presenteeism) [16]. As a result, the study also showed that higher perceived stress and higher rates of presenteeism among owners and managers at SMEs were associated with inefficiency and loss of productivity [16]. Hence, our findings indicate that, particularly owners and managers of MSE/SME settings would benefit from supporting offers to deal with these psychosocial stressors.

In the second discussion round, two main barriers were mentioned by the experts that could hinder the transfer of a comprehensive intervention to increase job satisfaction to other MSE/SME settings. The two main barriers were the following: (1) lack of access to MSE/SME settings to offer suitable approaches, and (2) that topics other than those chosen by the IMPROVE*job* intervention were considered as more important among owners, managers, and employees. In other studies, there are also some reports about how difficult it is to get access to MSE/SME settings, despite diverse recruitment strategies [17]. Reasons for the difficult access to MSE/SME settings are presumably that offers are not considered as relevant, or simply that the personnel and time resources are lacking. Other studies also reported the missing awareness of psychosocial topics. Pavlista et al. interviewed 18 owners and managers from 15 micro- and small-sized enterprises in Germany, and reported about a stigmatization of mental health [21]. As a result, the severity of psychosocial stress was not recognized among owners and managers, and it was often seen as a private or individual problem [21].

Based on the identified barriers, the experts developed various and detailed ideas for the transfer of the IMPROVE*job* intervention into other MSE/SME settings during the second discussion round. The experts’ ideas agreed in part with the drivers identified by Pavlista et al., which were the importance of easy access, the external support from experts, renaming terms, and showing the benefits of engaging in health promotion activities [21]. Easy access and low-threshold offers seem to be key for being able to address psychosocial topics in MSE/SME settings. The integration of external support, and experts with sufficient experience, is important, and can also increase the acceptance and discussion of psychosocial issues among employees [21]. Experts in our study emphasized the importance of “speaking the language” of MSE/SME settings. Pavlista et al. also reported that some owners and managers had negative associations with the term “psychosocial risk assessment”, and that it is therefore necessary to rename for example these terms [21]. Experts in our study also pointed out that the benefits must be clearly explained to the companies, so that they are motivated to address psychosocial issues at all. Pavlista et al. also hypothesize that, MSE/SME are more likely to address psychosocial issues when the personal relevance and importance for the organization has been clearly communicated and clarified [21], which is also shown by our study results.

Overall, it seems possible to transfer the multimodal participative IMPROVE*job* intervention from the GP to other MSE/SME settings, with the provision that, special characteristics and needs of MSE/SME settings are adequately considered and addressed. The transfer of the comprehensive participative IMPROVE*job* intervention to other MSE/SME settings requires therefore an adapted intervention format. The following main implications were derived for a possible adapted design of the IMPROVE*job* intervention into other MSE/SME settings: The separation into two workshops (leaders and leaders with teams) seems appropriate also for other MSE/SME settings. Participation in workshops should be made as easy as possible for the participants (e.g., regarding access, duration). The specific objectives and implementation of the two workshops should be determined in advance with owners and managers of the MSE/SME settings.The topics of the IMPROVE*job* intervention [30,33] seemed to be appropriate for other MSE/SME settings; however, they should be linked to topics that are currently relevant, such as a shortage of skilled workers. In general, it seems to be beneficial if the participants (owners/managers and their teams) themselves determine in advance which topics they would like to focus on during the workshops.The didactic parts of the IMPROVE*job* intervention could be expanded, with more involvement of skills labs, interdisciplinary exchanges, peer learning, and networking opportunities.The IMPROVE*job* facilitators should be available during the implementation period for further questions and problems, so that the IMPROVE*job* intervention can be carried out as planned. MSE/SME settings should be provided with special support during the implementation process.

In addition, the following general implications can further be derived for the design of WHM in MSE/SME settings:The implementation of psychosocial topics such as job satisfaction, leadership, teamwork, and social relationships, as well as work organization and work process should be further promoted in MSE/SME settings. Therefore, specific campaigns are useful to inform and raise awareness on these topics in MSE/SME settings.Due to limited personnel and time resources in MSE/SME settings, offers and campaigns with easy access are to be preferred. The Cardiff Memorandum on Workplace Health Promotion in SME settings, for example, identified limited resources as the main barrier for the implementation of workplace health promotion activities [34]. Another study also identified workload as the main barrier for participation in workplace health promotion offers [35]. Frequently, smaller companies have limited time and resources for promoting the well-being and health of employees; therefore, knowledge on workplace health promotion should be adapted to the needs of smaller companies [34].Smaller businesses differ from larger companies not only in their number of employees, structural organization, and financial resources, but also in terms of psychosocial experiences and the central impact of owners and managers [14]. Owners and managers in MSE/SME settings should therefore be supported and accompanied with regards to addressing psychosocial demands in their companies. External support could also be helpful, and should be available as a contact person for a longer period of time. In this context, the question should be discussed which persons, even outside of a study context, are available, and could provide support and raise awareness for WHM in MSE/SME settings. In our view, people with a professional background in occupational medicine would be particularly helpful here.Besides personal challenges for managers, general challenges like emerging topics, organizational issues, and challenges regarding team care and staff management should also be considered and addressed.It is important to provide an environment and a participative process where company owners, managers, and employees discuss the implementation of measures for the promotion and implementation of WHM on a long-term basis.

### Limitations and Strengths

Due to a very short timeframe for the overall project, the time for recruitment was limited (four weeks), and it was not possible to recruit employees from different companies to conduct further workshops exploring the views of employees. For example, the UK workplace stress survey revealed that employees in MSE/SME settings have lower stress levels than employees in larger companies, and it may be beneficial for employees to work in smaller companies [36]. So, our study involved only the perspective of experts in mainly leadership positions, and we could not capture the particular perspective of the employees. To obtain a more comprehensive picture, it is necessary to consider the perspectives of both leaders and employees of different hierarchies. Due to the ongoing COVID-19 pandemic, we conducted all workshops, including the focus group discussion and most of the single interviews, in a digital format or via telephone. Direct personal impressions and reactions from the experts are therefore widely missing, and could have turned out differently in a face-to-face situation.

Despite the limitations, our study also has several strengths. First, a broad spectrum of experiences, attitudes, and feedback was considered by the experts involved. The experts had various professional backgrounds and in-depth insights in MSE and SME settings, so that we received answers to our research questions, although the sample was small and cannot be considered as representative. But despite the small sample, we assume that we have reached a theoretical saturation of the data with regard to the experts interviewed. As outlined in the methods section, Malterud et al. described in a model, different items that should be considered during the recruitment process in a qualitative study [31]. According to the authors, fewer participants are needed for a qualitative study if the study objective is rather narrow, and the combination of participants is very specific to the study objective [31], as was the case in our study. Second, the inter- and transdisciplinary design enabled, in our opinion, a very beneficial exchange between science and practice. The study design was developed by the same interdisciplinary team that was substantially involved in the development of the IMPROVE*job* intervention. Therefore, we discussed the results also against the background of the developed IMPROVE*job* intervention. Furthermore, close interaction with the advisory board of experts during the transdisciplinary workshop, and the inclusion of additional experts for the qualitative data collection, allowed for the consideration of perspectives directly from the MSE/SME setting. The close participatory involvement of persons from the MSE/SME setting, and the discussion within the work group meetings, thus provide a basis for the further development of the IMPROVE*job* intervention in other MSE/SME settings.

## 5. Conclusions

This study represents a qualitative study that elaborates topics of the comprehensive multimodal participatory IMPROVE*job* intervention, and explores possible transfer options into other MSE/SME settings, with the consideration of the perspectives of different experts. The insights and perspectives of the experts from different MSE/SME settings were very helpful, and indicate what needs to be further considered when the IMPROVE*job* intervention is specifically adapted for possible use in other MSE/SME settings. Thus, the present study gives valuable information on how WHM in MSE/SME settings can be improved, not only with regard to psychosocial aspects, but also from a general perspective. Based on our study, further approaches and interventions to improve job satisfaction and reduce perceived stress and strain factors in other MSE/SME settings are crucially needed, and should be further developed. Some authors already recommend considering and including not only job satisfaction, but also other outcomes such as well-being of employees [37]. A tailored intervention in the context of WHM, and analogous to the IMPROVE*job* intervention, could be well suited to promote not only job satisfaction, but also the general well-being of employees [38]. Furthermore, it would be, in our opinion, useful and beneficial to evaluate an adapted version of the IMPROVE*job* intervention in future studies among owners, managers, and employees, to explore the potential application of this intervention in different MSE/SME settings.

## Figures and Tables

**Figure 1 ijerph-20-04067-f001:**
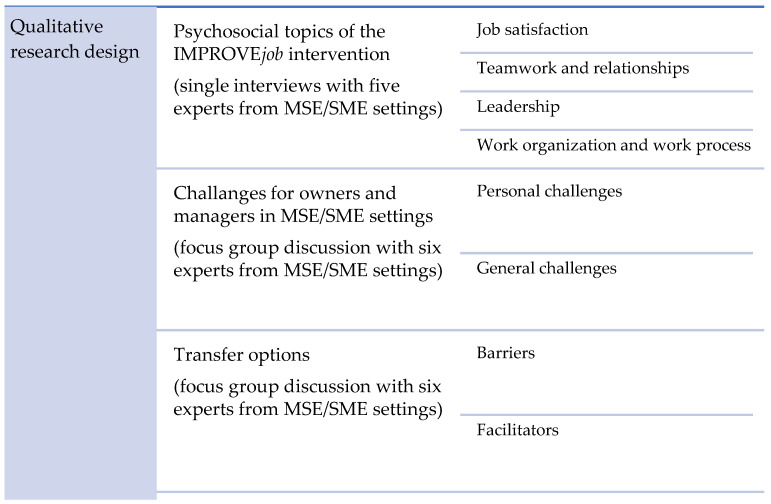
Overview of the results of the single interviews and the focus group discussion.

**Table 1 ijerph-20-04067-t001:** Overview of components of the IMPROVE*job* intervention [23,28,29].

Components	Aim
Workshop 1 (leaders only)	Addresses topics around leadership (leadership styles, leadership role conflicts, reflection opportunities, aspects of occupational health, and team care) targeting physicians with leadership responsibilities
Workshop 2 (leaders with teams)	Addresses topics around team communication, work organization, and work process, targeting physicians with leadership responsibilities and all practice employees; here, the teams decided which topics they wanted to work on over the course of the implementation period (9 months) to reduce psychosocial stress
Toolbox with supplemental material (for leaders and teams)	Supplemental material to consolidate the contents of the workshops afterwards (management logbook for physicians, logbook for practice staff, desk calendar for practice staff, additional material for download in a secured webspace)
IMPROVE*job* facilitators during a 9-month implementation period	Continuous accompaniment and support of the implementation period by on-site meetings and phone calls

**Table 2 ijerph-20-04067-t002:** Drivers for the promotion of job satisfaction, leadership, teamwork, and relationships, and work organization and work process.

	Job Satisfaction	Leadership	Teamwork and Relationships	Work Organization and Work Process
**Drivers for promotion**	Highlight individual performances within entire team (interview 1) Free scope for decision-making and time management (interview 1) Few controls (interview 1) Individually tailored work assignments (no under- or overstraining) (interview 1) Deployment according to preferences and potential (interviews 1–3) Expression of constructive criticism (Interview 2–3) and work-related requests (Interview 2) Consideration of personal life circumstances (interviews 3–4) High level of identification with company, product, customers, superiors (interview 5) Good working environment (interview 5) Personal contribution to the company’s success (interview 5) Constructive communication and leadership (interview 5)	Leadership as a “collective task” (interviews 1–3, interview 5) Promote cooperation and atmosphere in teams (interviews 2–3) Create good working conditions, foster employee health, ensure consistent compliance with occupational health and safety measures, and quality management (interviews 2–4) Promote professional development and training, and continuously work on changes in attitudes, behaviors, and experiences (interviews 2–3, interview 5) Workshops and individual coaching for managers (interview 5) Influence on sufficient time of employees to complete their work tasks (interview 1) Exchange and reflection with other leaders (interview 5)	Good exchange and clarification of unpleasant issues (interview 1) Definition/identification of common goals (interview 3) Team is responsible for something (e.g., project, task, customer), knows objective and own area of responsibility, as well as contribution to team performance (interview 1, interview 5) Actions to get to know each other better (interview 1) Consideration of suitability for team when selecting new employees (interview 1) Supporting exchange and joint activities (interview 1)	Sufficient communication, precise assignment of tasks and work description per order (interview 1) Clear responsibilities, clearly defined work tasks, coordinated customer assignment (interviews 1–3) Efficient project and schedule management (order acceptance, deadlines) (interviews 2–4) Short decision-making paths (interview 2) Process documentation: how do processes run (interview 5)

**Table 3 ijerph-20-04067-t003:** Facilitators for the transfer of the topics of the IMPROVE*job* intervention into other MSE/SME settings.

How to raise interest and contact different MSE/SME settings	Address a relevant general topic for MSE/SME settings (e.g., shortage of skilled workers), and introduce additional topics (e.g., aspects of leadership, implementation of psychosocial risk assessments) by emphasizing the close interconnection between them Promote motivation for WHM (understand WHM as a comprehensive concept, not as singular measures) Try voluntary paths for occupational health and safety topics (e.g., as a means to increase productivity, intrinsic motivation), no use of legal obligations and external pressure (“no threat of occupational health and safety”) No sanctioning from the top Short introductory workshops targeted at relevant topics for MSE/SME settings (e.g., strengthen resilience for owners, managers, and employees; if requested, arrangement of additional workshops or provision of specified material “Speak the language of MSE/SME settings” and identify needs. Communicate benefits for MSE/SME settings (e.g., “Want to solve real problems in MSE/SME”, “nice to have” is not enough) Cost-benefit analysis, and remove obstacles and burdens together Include, involve, and support owners and managers (owners and managers should act as initiators), and experts “Nagging long enough”
Methodical facilitators to implement relevant topics in MSE/SME settings	Promotion of learning and experimentation spaces (cross-sector exchange) Company breakfasts to consolidate social relations and a culture of professional exchange in a relatively informal setting Training (e.g., aspects of communication and leadership) in skills labs combining theoretical and practical impulses Peer to peer approach, and peer learning Networking, and spaces for open exchange (e.g., online exchange, address MSE/SME networks) Regular, short-term offers (easy access without registration, “sniffing”, not feeling obligated, possibility of contacting speakers afterwards if needed) Integrate health, prevention, and resilience as “must have” in training programs (“positive branding”) Create best practice guidelines (e.g., family-friendly company) meeting employees’ work, family, and well-being needs Consider different structures and working methods in middle management at MSE/SME settings

## Data Availability

With regards to publication of personal information: the collected data are not available.

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
