# Peer review of "Exploring Transfer Potentials of the IMPROVEjob Intervention for Strengthening Workplace Health Management in Micro-, Small-, and Medium-Sized Enterprises in Germany: A Qualitative Study"

_ijerph, 2023, doi:10.3390/ijerph20054067_

Round 1

Reviewer 1 Report (Previous Reviewer 2)

I am pleased with the new version

Author Response

Reviewer 2 Report (New Reviewer)

I would recommend this article to be titled -Part 1. As authors have written in  a section 4.1 (475-476) -"it is short time for the survey (4 weeks), and it was not possible to recruit employees from different companies  to conduct further workshops exploring the views of employees." 

The study involves only the opinions of experts mainly at leadership positions, and doesn't present the  perspective of the employees. To obtain a more comprehensive picture, it is necessary to consider the perspectives  of both leaders and employees of different hierarchies in article Part 2. 

Next article to be focused more on specifics in Factors for job satisfaction in each sector where SMEs are functioning.

Author Response

Reviewer 3 Report (New Reviewer)

Dear Authors,

Please consider my recommendations below:

1. the title is too long. I suggest to shorten it by using maximum 3 key phrases so as to increase the visibility in the scientific databases

2. please, revise English with a native speaker.

3. a figure showing the research methodology, the relationships among its parts and number of participants would be really helpful for the reader

4. the data analysed are retrieved from very small samples which make all the research non-representative (qualitative research samples, even if generally non-representative, shall consist of 15-20 subjects)

Yours faithfully,

Round 2

Reviewer 3 Report (New Reviewer)

Dear Authors,

I agree with the publication of your manuscript. 

Yours faithfully,

This manuscript is a resubmission of an earlier submission. The following is a list of the peer review reports and author responses from that submission.

Round 1

Reviewer 1 Report

I am honored to have the opportunity to review this research. And it is an interesting study to be concerned about in Psychosocial stressors in micro, small, and medium-sized enterprises. I have some comments for the author.

Q1: In the literature, the important literature of this study has been written, but the research object is German SMEs. In MSE/SME settings, health management is often not well implemented. Can the reasons be presented in more detail?

Q2. The research design shows the detailed research process and content, However, the recruitment method of experts can be described in detail ?

Q3. The research department is conducted in the form of expert group discussions in qualitative research. How to deal with the reliability and validity issues in qualitative research?

Q.3In addition to improving the work health management environment and improving job satisfaction, this study can further explain how to increase employees' well-being.

Reviewer 2 Report

I read the article with great interest. Source notes indicate that this is an interesting topic and current.

From the theoretical and practical point of view, all the elements used so far were thoroughly introduced. Research part - impressive. Very good descriptions, all legible and clear.

Overall, I am very impressed with the work. It is orderly and pleasant to read.

Reviewer 3 Report

Thanks for the opportunity to read this article. It deals with an interesting topic. However, I found several concerns that jeopardize its publication. I think it needs to be totally reshuffled in order to be properly finished as the authors will need to deal with the structure.

The introduction is totally biased (and conditioned) by the project they analyzed. As a result, the RQs are related to the project and not to the problems the article seeks to address. As such, the introduction needs to be simplified and the project needs to be presented in the methods section as a case study (to deal with the RQs and objectives) in order not to condition the RQs and conclusions.

The paper lacks a proper review of the literature where the authors manage to discuss the main aspects of workplace health management, the psychological stressors and the particularities of MSEs and SMEs. As such, the paper links the introduction to the methods section. As a result, the paper is orphan of a proper literature review on the subjects.

The discussion section of the IMPROVEjof intervention project starts with literature added during the discussion that was not even introduced in the literature review. See for example [31]. Moreover, it is a systematic literature review that should have been introduced right at the beginning of the paper!.

The authors seem to do a literature review during the discussion. See for example lines 393-397 in which the content and detail of study [10] is provided for the first time in the discussion. The sam occurs in lines 404-408. This means that the paper is ill-structured. The literature has to be presented first.

I would recommend the authors reshuffle completely the study, prepare a proper introduction and literature review in different sections and not to mix the discussion with a literature review. Complementarily, I would recommend thinking on the contribution and conversation of the reality and then to fit the project to the reality and see the added value and not as it is in which they converse about the project and try to fit the reality to the project.